# Using neural networks to improve simulations in the gray zone

Raphael Kriegmair[1], Yvonne Ruckstuhl[1], Stephan Rasp[2], and George Craig[1]

[1]Meteorological Institute Munich, Ludwig-Maximilians-Universität München, Germany
[2]ClimateAi, Inc.

**Correspondence:** Yvonne Ruckstuhl (yvonne.ruckstuhl@lmu.de)

**Abstract.** Machine learning represents a potential method to cope with the gray zone problem of representing motions in dynamical systems on scales comparable to the model resolution. Here we explore the possibility of using a neural network to directly learn the error caused by unresolved scales. We use a modified shallow water model which includes highly nonlinear processes mimicking atmospheric convection. To create the training dataset we run the model in a high and a low-resolution setup and compare the difference after one low resolution time step starting from the same initial conditions, thereby obtaining an exact target. The neural network is able to learn a large portion of the difference when evaluated on single time step predictions on a validation dataset. When coupled to the low-resolution model, we find large forecast improvements up to one day on average. After this, the accumulated error due to the mass conservation violation of the neural network starts to dominate and deteriorates the forecast. This deterioration can effectively be delayed by adding a penalty term to the loss function used to train the ANN to conserve mass in a weak sense. This study reinforces the need to include physical constraints in neural network parameterizations.

## 1 Introduction

Current limitations on computational power force weather and climate prediction to use relatively low resolution simulations. Subgrid scale processes, i.e. processes that are not resolved by the model grid, are typically represented using physical parameterizations (Stensrud, 2009). Inaccuracies in these parameterizations are known to cause errors in weather forecasts and biases in climate projections. While parameterizations are becoming more sophisticated over time, there is evidence that key structural uncertainties remain (Randall et al., 2003; Randall, 2013; Jones and Randall, 2011).

A particularly difficult problem in the representation of unresolved processes is the so-called gray zone (Chow et al., 2019; Honnert et al., 2020), where a certain physical phenomenon such as a cumulus cloud is similar in size to the model resolution and hence partially resolved. In the development of many classical parameterizations, features are assumed to be small in comparison to the model resolution. This scale separation provides a conceptual basis for specifying the average effects of the unresolved flow features on the resolved flow. In contrast, there is no theoretical basis for determining such a relationship in the gray zone. Instead, the truncation error of the numerical model is a significant factor. While we might still expect there to be some relationship between the resolved and unresolved parts of the flow, we have no way to define it.

Viewing the atmosphere as a turbulent flow, with up- and downscale cascades, phenomena like synoptic cyclones and cumulus clouds emerge where geometric or physical constraints impose length scales on the flow (Lovejoy and Schertzer, 2010;

Marino et al., 2013; Faranda et al., 2018). If a numerical model is truncated near one of these scales, the corresponding phenomenon will be only partially resolved and the simulation will be inaccurate. In particular, the properties of the phenomenon may be determined by the truncation length, rather than by the physical scale. A thorough review of the gray zone problem from a turbulence perspective is provided by Honnert et al. (2020).

An important example of the gray zone in practice is the simulation of deep convective clouds in kilometer-scale models used operationally for regional weather prediction. The models typically have a horizontal resolution of 2-4 km, which is not sufficient to fully resolve the cumulus clouds with sizes in the range from 1 to 10 km. In these models, the simulated cumulus clouds collapse to a scale proportional to the model grid length, unrealistically becoming smaller and more intense as resolution is increased (Bryan et al., 2003; Wagner et al., 2018). In models with grid lengths over 10 km, the convective clouds are completely subgrid and should be parameterized, while models with resolution under 100 m will accurately reproduce the dynamics of cumulus clouds provided that the turbulent mixing processes are well represented. In the gray zone in between, the performance of the models depends sensitively on resolution and details of the parameterizations that are used (Jeworrek et al., 2019).

Using machine learning methods such as artificial neural networks (ANNs) for alleviating the problems described above has received increasing attention over the past years. One approach is to avoid the need of parameterizations all together by emulating the entire model using observations (Brunton et al., 2016; Pathak et al., 2018; Faranda et al., 2020; Fablet et al., 2018; Scher, 2018; Dueben and Bauer, 2018). In these studies a dense and noise free observation network is often assumed. Brajard et al. (2020a) and Bocquet et al. (2020) circumvent the requirement of this assumption by using data assimilation to form targets for ANNs from sparse and noisy observations.

Though studies have shown that surrogate models produced by machine learning can be accurate for small dynamical systems, replacing an entire numerical weather prediction model for operational use is not yet within our reach. Therefore, a more practical approach is to use ANNs as replacement for uncertain parameterizations. This has been done either by learning from physics based expensive parametrization schemes (O'Gorman and Dwyer, 2018; Rasp et al., 2018) or high resolution simulations (Krasnopolsky et al., 2013; Brenowitz and Bretherton, 2019; Bolton and Zanna, 2019; Rasp, 2020; Yuval and O'Gorman, 2020), which is the approach we take here. Such data driven techniques could be a way to reduce the structural uncertainty of traditional parameterizations, even at gray zone resolutions where the physical basis of the parameterization is no longer valid. The first challenge is to create the training data, i.e. to separate the resolved and unresolved scales from the high-resolution simulation. Brenowitz and Bretherton (2019) use a coarse-graining approach based on subtracting the coarse grained advection term from the local tendencies. This approach can be used for any model and resolution but is sensitive to the choice of grid and time step. Further, the resulting subgrid tendencies are only an approximation and may not represent the real difference between the low and high-resolution model. Yuval and O'Gorman (2020) use the same model version for low and high-resolution simulations and compute exact differences after a single low-resolution time step by starting both model versions from the same initial conditions. They manage to obtain stable long-term simulations using the low-resolution model with a machine learning correction that come close the the high-resolution ground truth.

Here, we use the modified rotating shallow water (modRSW) model to explore the use of a machine learning subgrid representation in a highly non-linear dynamical system. The modRSW is an idealized fluid model of convective-scale numerical weather prediction, in which convection is triggered by orography. As such, the model mimics the gray zone problem of operational kilometer-scale models. Using a simplified model allows us to focus on some key conceptual questions surrounding machine learning parameterizations, such as how choices in neural network training affect long-term physical consistency. In particular, we include weak physical constraints in the training procedure.

The contents of this work are outlined in the following. Section 2 introduces the experiment setup used to obtain and analyze results. The modRSW model is briefly explained in Section 2.1, followed by a description of the training data generation in Section 2.2. The architecture and training process of the ANN used in this research are given in 2.3. Our verification metrics are defined in Section 3. The results are presented in Section 4, followed by concluding remarks in Section 5.

## 2   Experiment setup

### 2.1   The modRSW Model

The modRSW model (Kent et al., 2017) used in this research represents an extended version of the 1D shallow water equations, i.e. 1D fluid flow over orography. Its prognostic variables are fluid height $h$, wind speed $u$ and a rain mass fraction $r$. Based on the model by Würsch and Craig (Würsch and Craig, 2014) it implements two threshold heights, $H_c < H_r$, initiating convection and rain production, respectively. Convection is stimulated by modifying the pressure term to remain constant where $h$ rises above $H_c$. In contrast to Würsch and Craig (2014), the modRSW model does not apply diffusion or stochastic forcing. The model is mass conserving, meaning that the domain mean of $h$ is constant over time. In this study, a small but significant model-intrinsic drift in the domain mean of $u$ is removed by adding a relaxation term. This term is defined using a corresponding time scale $t_{relax}$, as $(\bar{u}_0 - \bar{u}_t) \cdot t_{relax}$, where the overbar denotes the domain mean. Depending on the orography used, this model yields a range of dynamical organization between regular and chaotic behaviour. Orography is defined as a superposition of cosines with wavenumbers $k = 1/L, ..., k_{max}/L$ (L domain length). Amplitudes are given as $A(k) = 1/k$, while phase shifts for each term are randomly chosen from $[0, L]$. In this work, two realizations of the orography are selected to represent regular and more chaotic dynamical behavior. Figure 1 displays a 24 hour segment of the simulation corresponding to each orography.

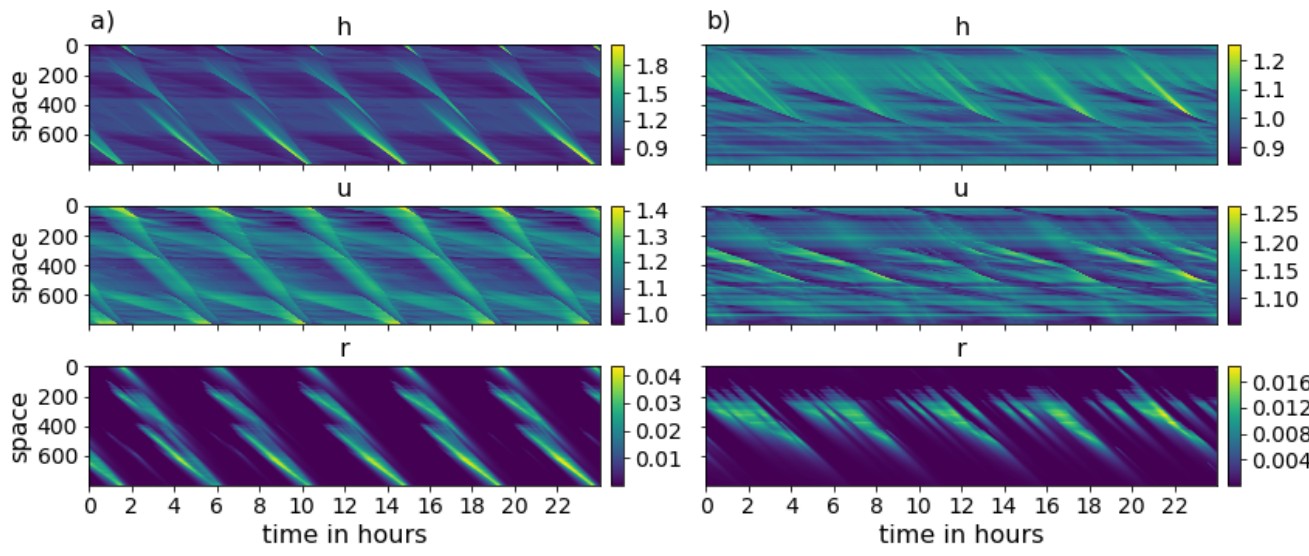

**Figure 1.** 24 hour segment of the HR simulation for the three model variables $h$, $u$, $r$ (from top to bottom) corresponding to the regular case (left panel) and the chaotic case (right panel).

## 2.2 Training Data Generation

Conceptually, the ANN's task is to correct a low resolution (LR) model forecast towards the model truth, which is a coarse grained high resolution (HR) model simulation. The coarse graining factor in this study is set to 4, which is analogous to the range of scales found in the gray zone where deep cumulus convection is partially resolved (e.g. 2.5-10 km). Faranda et al. (2020) show that the choice of coarse graining factor can substantially affect the performance of ML methods. In our case, however, choosing a larger factor would correspond to a coarse model grid length that is larger than the typical cloud size, changing the nature of the problem from learning to improve poorly resolved existing features in the coarse simulation to parameterizing features that might not be seen at all. The dynamical time step of the model is determined at each iteration based on the Courant–Friedrichs–Lewy (CFL) criterion. To achieve temporally equidistant output states for both resolutions, the time step is truncated accordingly when necessary.

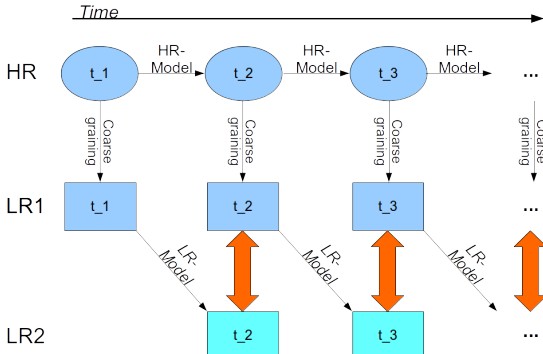

**Figure 2.** Schematic of training data generation process. A HR run is coarse grained to LR to generate model truth. Each model truth state is integrated forward for one time step using LR dynamics. The difference between the obtained states and corresponding model truth defines the desired network output (red arrows), while the preceding model truth defines the network input.

A training sample (input-target pair) is defined by the model truth at some time $t_i$ and the difference between the model truth and the corresponding LR forecast at $t_{i+1} = t_i + dt$, respectively (see Figure 2). To generate the model truth, HR data are obtained by integrating the modRSW model forward using the parameters shown in Table 1. All states and the orography
are subsequently coarse grained to LR, resulting in model truth (LR1). Each LR1 state is integrated forward for a single timestep using the modRSW model on LR with the coarse grained orography, resulting in a single step prediction (LR2). The synchronized differences $LR1(t_i) - LR2(t_i)$ then define the training targets corresponding to the input $LR1(t_{i-1})$, which includes the orography. A time series of $T = 200000$ time steps, which is equivalent to approximately 57 days in real time, is generated for both orographies. The first day of the simulation is discarded as spin up, the subsequent 30 days are used for
training and the remaining 26 days are used for validation purposes. The decorrelation length scale of the model is roughly 4 hours.

### 2.3 Convolutional ANN

A characteristic property of convolutional ANNs is that they reflect spatial invariance and localization. These two properties also apply to the dynamics of many physical systems, such as the one investigated here. They differ from e.g. dense networks
by the use of a so called kernel. This vector is "moved" step by step across the domain grid, covering $k$ grid points at each position. At each position, the dot product of kernel and current grid values is computed, determining (along with an activation function) the corresponding output value. For more details on convolutional ANNs we refer to Goodfellow et al. (2016).

The ANN structure used in this research is described in the following. 5 hidden layers are applied, each using the *ReLU* activation function. The input layer uses *ReLU* as well, while the output layer uses a linear activation function. All hidden
layers have 32 filters. The input and output layer shapes are defined by input and target data. The kernel size is set uniformly to 3 grid points.

| Model Parameter | Symbol | Value | Notes |
|---|---|---|---|
| HR gridpoint number | $N_{HR}$ | 800 | - |
| LR gridpoint number | $N_{LR}$ | 200 | - |
| Time step | $dt$ | 0.001 | - |
| Domain size (non dim.) | $L$ | 1.0 | - |
| CFL | - | 0.5 | - |
| Convection threshold | $H_c$ | 1.02 | - |
| Rain threshold | $H_r$ | 1.05 | - |
| Initial total height | $H_0$ | 1.0 | - |
| Rossby Number | $Ro$ | $\infty$ | - |
| Froude Number | $Fr$ | 1.1 | - |
| Effective gravity | $g$ | $Fr^{-2}$ | - |
| Beta | $\beta$ | 0.2 | - |
| Alpha | $\alpha^2$ | 10 | - |
| Rain Conversion Factor | $c^2$ | $0.1 \times g \times H_r$ | - |
| Wind Relaxation time scale | $t_{relax}$ | $dt$ | - |
| **Orography Generation** | | | |
| Maximum wave number | $k_{max}$ | 100 | - |
| Maximum Amplitude | $B_{max}$ | 0.1 | - |

**Table 1.** Model setting parameters

The loss is determined during training by comparing the ANN output to the corresponding target. A standard measure for loss is the mean squared error (MSE). However, any loss function can be used to tailor to the application. For example, additional terms can be added to impose weak constraints on the training process, as for example done in Ruckstuhl et al. (2021). This possibility is exploited here to impose mass conservation in a weak sense. The constraint is implemented by penalizing the deviation of the square of the domain mean corrections of *h* from zero, resulting in the following loss function

$$\text{MSE}(y_{out}, y_{target}) + w_{mass} \cdot \left( \overline{y_{out}^h} \right)^2 \tag{1}$$

where the second term represents a weighted mass conservation constraint. In this expression, $y_{out}$ and $y_{target}$ are the output and corresponding target of the ANN respectively, MSE denotes the mean squared error, the tunable scalar $w_{mass}$ is the mass conservation constraint weighting, $y_{out}^h$ is the ANN output for *h* and the overbar denotes the domain mean.

The Adam algorithm with a learning rate of $10^{-3}$ is used to minimize the loss function over the ANN weights in batches of 256 samples. Since the loss function is typically not convex, the ANN likely converges to a local rather than the global minimum. To sample this error, we repeat the training of each ANN presented in this paper with randomly chosen initial

weights 5 times. For all ANNs a total of 1000 epochs is performed. The ANN architecture and hyperparameters were selected based on a loose tuning procedure, where no strong sensitivities were detected. The training is done using python library Keras (Chollet et al., 2015).

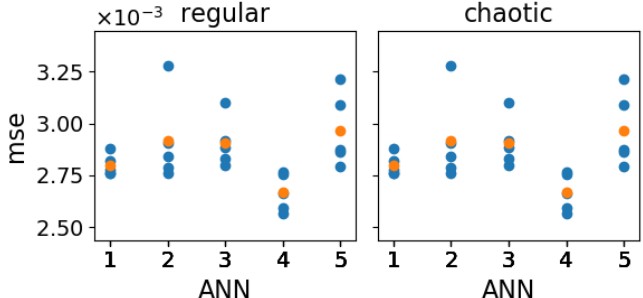

**Figure 3.** Loss function value for $w_{mass} = 0$ (mse) of the validation data corresponding to the last 5 epochs of the training process (blue) for each trained ANN (x-axis). For each ANN the mean loss function value over the last 5 epochs is depicted in orange.

## 3   Verification methods

As the initial training weights of the ANNs and the exact number of epochs performed is to some extent arbitrary, it is desirable to measure the sensitivity of our results to the realization of these quantities. Figure 3 shows the MSE of the validation data set of the last 5 epochs (y-axis) for 5 ANNs with different realizations of initial training weights (x-axis) for both orographies. Since the MSE appears sensitive to both the initial weights and the epoch number, we use both to sample the total ANN variability, resulting in 5×5=25 samples for each ANN training setup that is presented in the remainder of this paper.

The main scores that are used to verify the efficacy of the ANNs are the root mean squared error (RMSE), spatial mean error (SME) and bias:

$$\text{RMSE}(\mathbf{y}) = \sqrt{\frac{1}{N_{LR}} \sum_{j=1}^{N_{LR}} \left(y_j - y_j^{true}\right)^2} \tag{2}$$

$$\text{SME}(\mathbf{y}) = \left| \frac{1}{N_{LR}} \sum_{j=1}^{N_{LR}} \left(y_j - y_j^{true}\right) \right| \tag{3}$$

$$\text{bias}(\mathbf{y}) = \frac{1}{N_{LR}} \sum_{j=1}^{N_{LR}} \left(y_j - y_j^{true}\right) \tag{4}$$

where $N_{LR} = 200$ is the number of grid points and $\mathbf{y}^{true} \in \mathbb{R}^{N_{LR}}$ is a snapshot of the model truth. Multiple samples of these scores are obtained using both the 25 realizations of the ANN, and a sequence of initial conditions provided by the

time dimension. The final verification metrics are then the mean and standard deviation (SD) of the respective scores $X \in$ {RMSE, SME, bias}:

$$\overline{X}(\mathbf{y}) = \frac{1}{L_{ANN}T_{veri}} \sum_{l=1}^{L_{ANN}} \sum_{t=0}^{T_{veri}} X(\mathbf{y}_t^l) \tag{5}$$

$$\text{SD}_{\text{total}}\left(X(\mathbf{y})\right) = \sqrt{\frac{1}{L_{ANN}T_{veri}} \sum_{l=1}^{L_{ANN}} \sum_{t=0}^{T_{veri}} \left(X(\mathbf{y}_t^l) - \overline{X}(\mathbf{y})\right)^2} \tag{6}$$

$$\text{SD}_{\text{time}}\left(X(\mathbf{y})\right) = \frac{1}{L_{ANN}} \sum_{l=1}^{L_{ANN}} \sqrt{\frac{1}{T_{veri}} \sum_{t=0}^{T_{veri}} \left(X(\mathbf{y}_t^l) - \overline{X}(\mathbf{y}^l)\right)^2} \tag{7}$$

$\quad$ $$\text{SD}_{\text{ANN}}\left(X(\mathbf{y})\right) = \frac{1}{T_{veri}} \sum_{l=0}^{T_{veri}} \sqrt{\frac{1}{L_{ANN}} \sum_{l=1}^{L_{ANN}} \left(X(\mathbf{y}_t^l) - \overline{X}(\mathbf{y}_t)\right)^2} \tag{8}$$

where $t$ and $l$ index time and ANN realizations respectively, $\overline{X}(\mathbf{y}_t)$ indicates the mean over time steps and $\overline{X}(\mathbf{y}^l)$ the mean over ANN realizations. Note that equations (7) and (8) are meant to isolate the variability inherited from initial conditions and ANN realizations respectively. We apply these verification metrics to both single time step predictions and 48-hour forecasts.

- **Single time step predictions:**

$\quad$ For each time step corresponding to the validation data set ($T_{veri} = 92863$), the model truth is used as initial condition for a single time step prediction of the LR model, creating a LR prediction ($LR$). This LR prediction is subsequently corrected by the ANN, creating the corresponding ANN-corrected prediction ($LR_{ANN}$).

- **48-hour forecasts:**

$\quad$ The 48-hour forecasts are generated from a set of 50 initial conditions ($T_{veri} = 50$) taken from the validation data set. To
165 $\quad$ ensure independence, the initial conditions are set 4 hours apart, which is roughly the decorrelation length scale of the model. After each low resolution single time step prediction, the ANN is applied to create initial conditions for the next LR single time step prediction, creating a 48-hour LR ANN-corrected forecast ($LR_{ANN}$). As reference, a LR simulation without ANN corrections ($LR$) is run in parallel.

For the single time step predictions and the 48-hour forecasts our verification metrics are applied to both $LR$ and $LR_{ANN}$ and
170 $\quad$ compared in section 4. Note that $L_{ANN} = 1$ for $LR$, yielding $\overline{X}(LR) = \text{SD}_{\text{ANN}}(X(LR))$ and $\text{SD}_{\text{total}}(X(LR)) = \text{SD}_{\text{time}}(X(LR))$.

## 4 Results

We performed a series of experiments designed to investigate the feasibility of using an ANN to correct for model error due to unresolved scales. In section 4.1 we first explore the performance of the ANNs trained with the standard MSE as loss function

$(w_{mass} = 0$ in equation (1)). Next, the weak constraint is added to the loss function as in equation (1) and the benefits are examined in section 4.2.

## 4.1 ANN with standard loss function

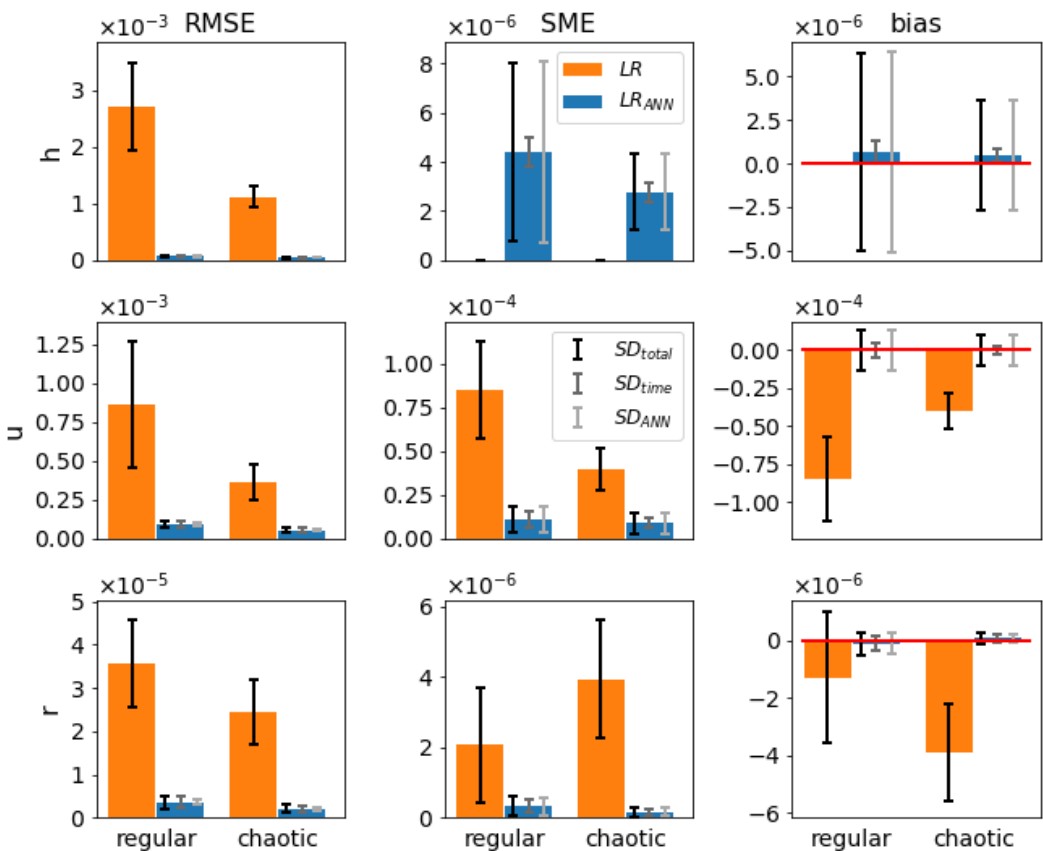

**Figure 4.** Mean (bars) and standard deviations $SD_{total}$, $SD_{time}$, $SD_{ANN}$ (error bars, from dark to light respectively) of the RMSE (left panel), SME (middle panel) and bias (right panel) of ANN corrected (blue) and uncorrected (orange) single time step predictions of the validation data with respect to the model truth for the regular and chaotic case (y-axis) and for variables $h$, $u$, $r$ (from top to bottom).

Figure 4 shows the results for the single time step predictions. The improvements achieved by the ANN with respect to $LR$ in terms of RMSE are substantial, for $h$,$u$ and $r$ amounting to 97%, 89% and 90% for the regular case and 96%, 84% 92% for the chaotic case. Also, the negative biases present in $u$ and $r$ for $LR$ virtually disappear when the ANN is applied. However, as the ANN is not explicitly instructed to conserve mass, a small SME is introduced in variable $h$. Its significance will become apparent when analyzing the 48-hour forecasts. It is interesting to note that the ANN's architecture and learning process is unbiased (small $\overline{bias}$), although single ANN realizations may be biased (large $SD_{ANN}$). Also, in contrast to the SME and bias,

for the RMSE the initial conditions are the main source of variability ($SD_{time} \gg SD_{ANN}$). This is better visible in Figures 7 and 8, where the results for $w_{mass} = 0$ are plotted again.

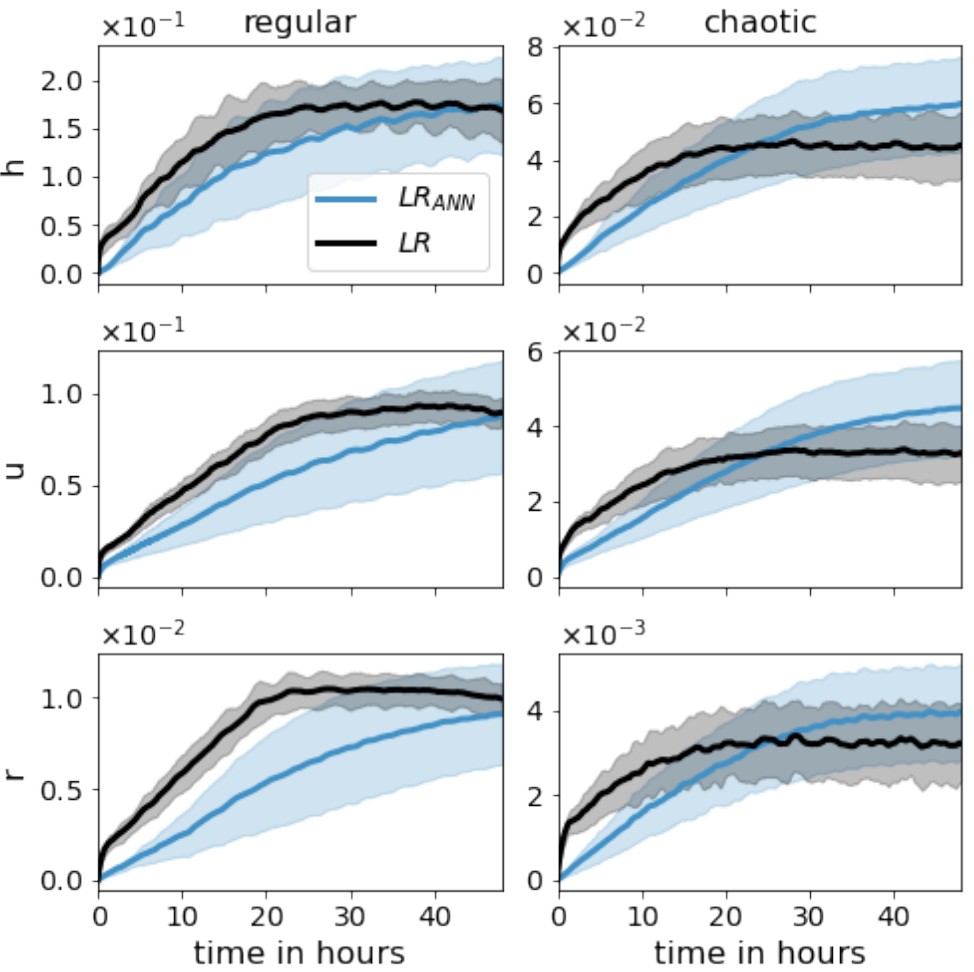

**Figure 5.** $\overline{\text{RMSE}}$ evolution of 48-hour forecasts for model variables $h$, $u$ and $r$ (from top to bottom) of $LR$ (black) and $LR_{ANN}$ (blue). The shaded region corresponds to $SD_{total}$.

Next we examine the effect of the ANN on a 48-hour forecast. Here we compare the LR simulation with ($LR_{ANN}$) and without ($LR$) the use of the ANN. Both simulations start from the same initial conditions as the model truth. The evolution of $\overline{\text{RMSE}}$ is presented in Figure 5. The RMSEs corresponding to the regular case are higher than for the chaotic case. This is because the regular case exhibits a repeating pattern of long-lived, high amplitude convective events. In comparison, the chaotic case produces short-lived perturbations with very small amplitude, leading to smaller climatological variability.

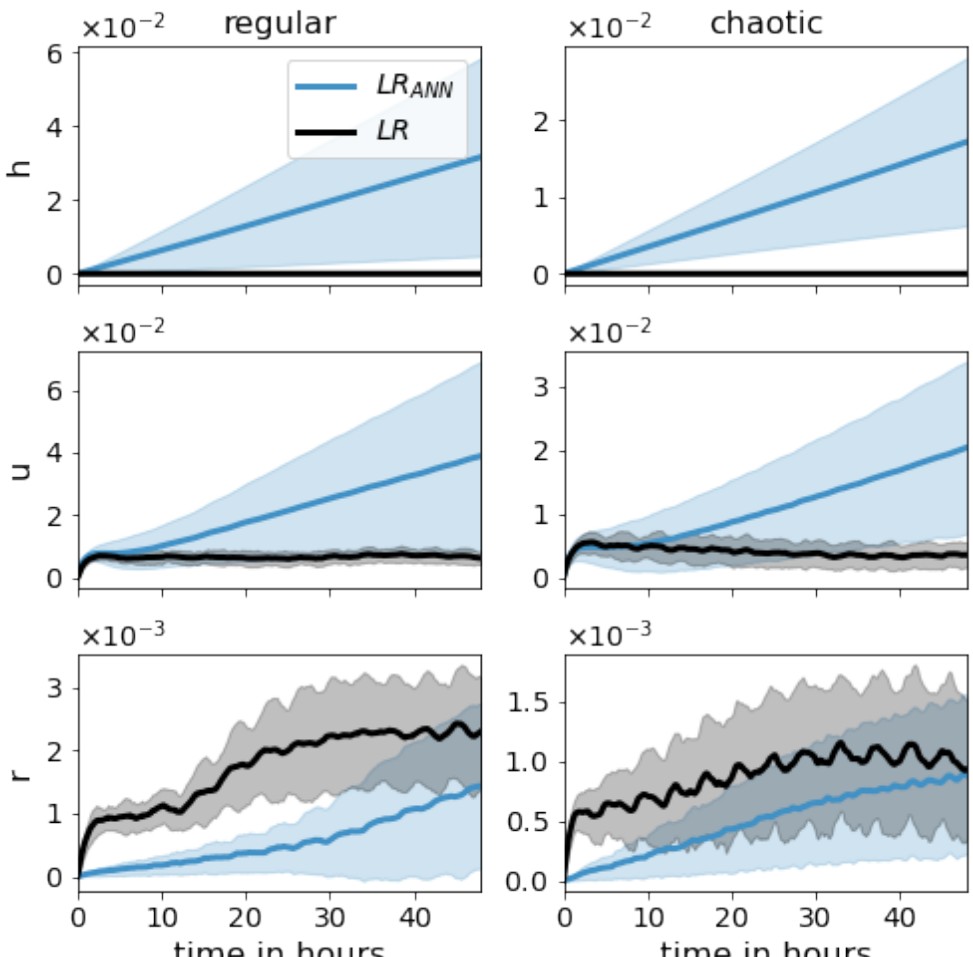

**Figure 6.** Same as Figure 5, but for the MSE.

For both orographies the ANN has a clear positive effect on the forecast until the error of $LR$ saturates, after which the error of $LR_{ANN}$ continues to grow. For the chaotic case this leads to a detrimental impact of the ANN after about 1 day. Also, the $SD_{total}$ of $LR_{ANN}$ is rapidly exceeding that of $LR$. This is because, in contrast to $LR$, the shaded region for $LR_{ANN}$ includes the variability due to the ANN realizations which significantly contributes to the total variability. This is seen in Figure 12 and discussed further in the next section.

It is not surprising that $LR_{ANN}$ deteriorates as the forecast lead time increases, since the ANNs are not perfect (as opposed to the data they were trained on) and the resulting errors accumulate over time, leading to biases. This is clearly visible in Figure 6, where it is seen that the SME of $h$ and $u$ diverge, in contrast to $LR$. The SME of $r$ for $LR$ is the result of a negative bias in the amount of rain produced (see Figure 4), caused by the coarse graining of the orography. This bias is significantly

reduced by the ANNs. The divergence of the SME of $h$ is the result of applying ANNs that, in contrast to the model, do not

conserve mass. This leads to accumulated mass errors, causing biases in the wind field due to momentum conservation and a change in probability for the fluid to rise above $H_c$ and $H_r$. We therefore investigate if reducing the mass error, by adding a penalty term to the loss function of the ANN, can increase the forecast skill further.

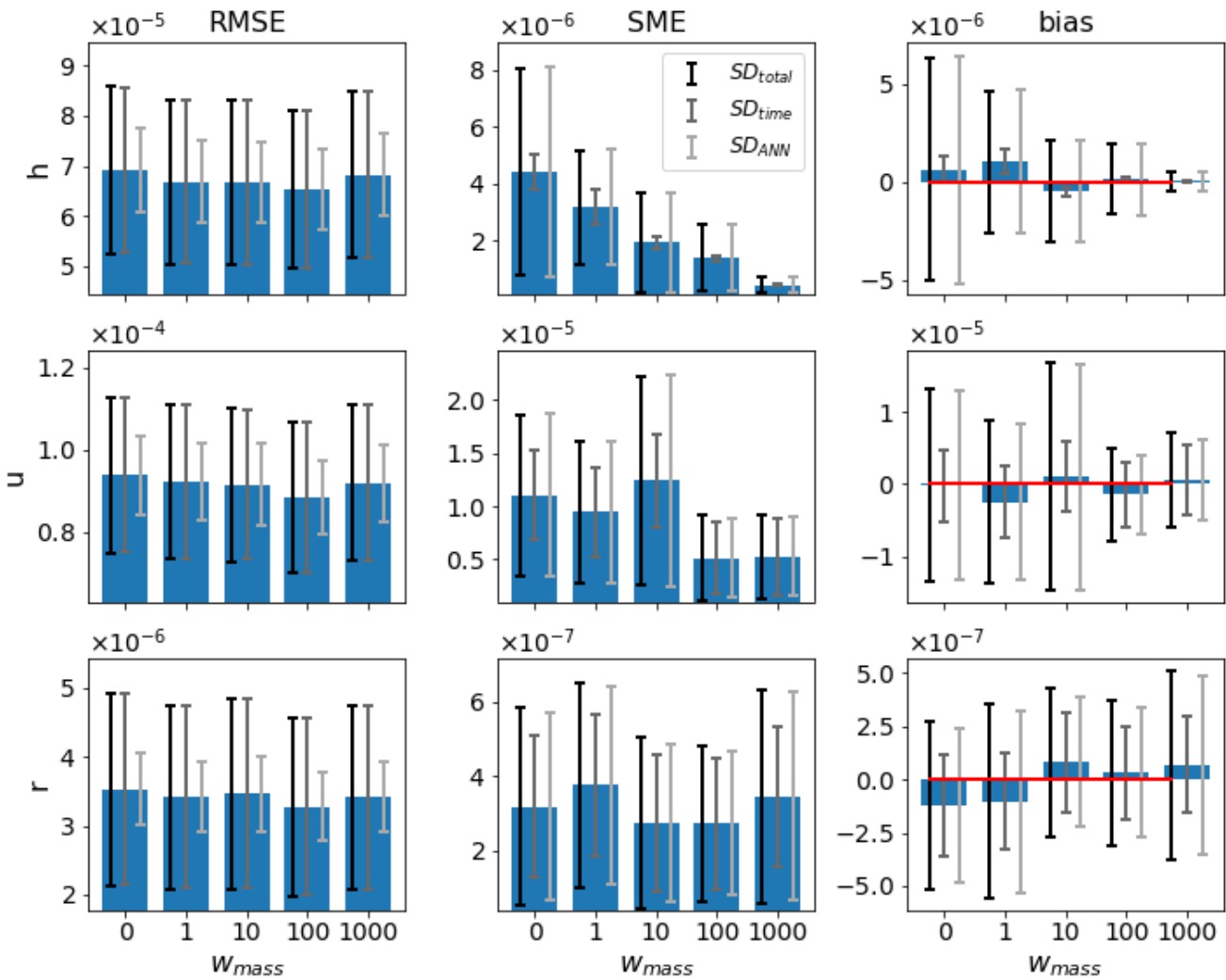

**Figure 7.** $\overline{\text{RMSE}}$ (left), MSE (middle), and bias (right) of the validation data for the different weightings (x-axis) and the respective model variables (rows) for the regular case. Error bars indicate (from dark to light) $\text{SD}_{\text{total}}$, $\text{SD}_{\text{time}}$ and $\text{SD}_{\text{ANN}}$ and the red lines in the right panel indicate the zero line.

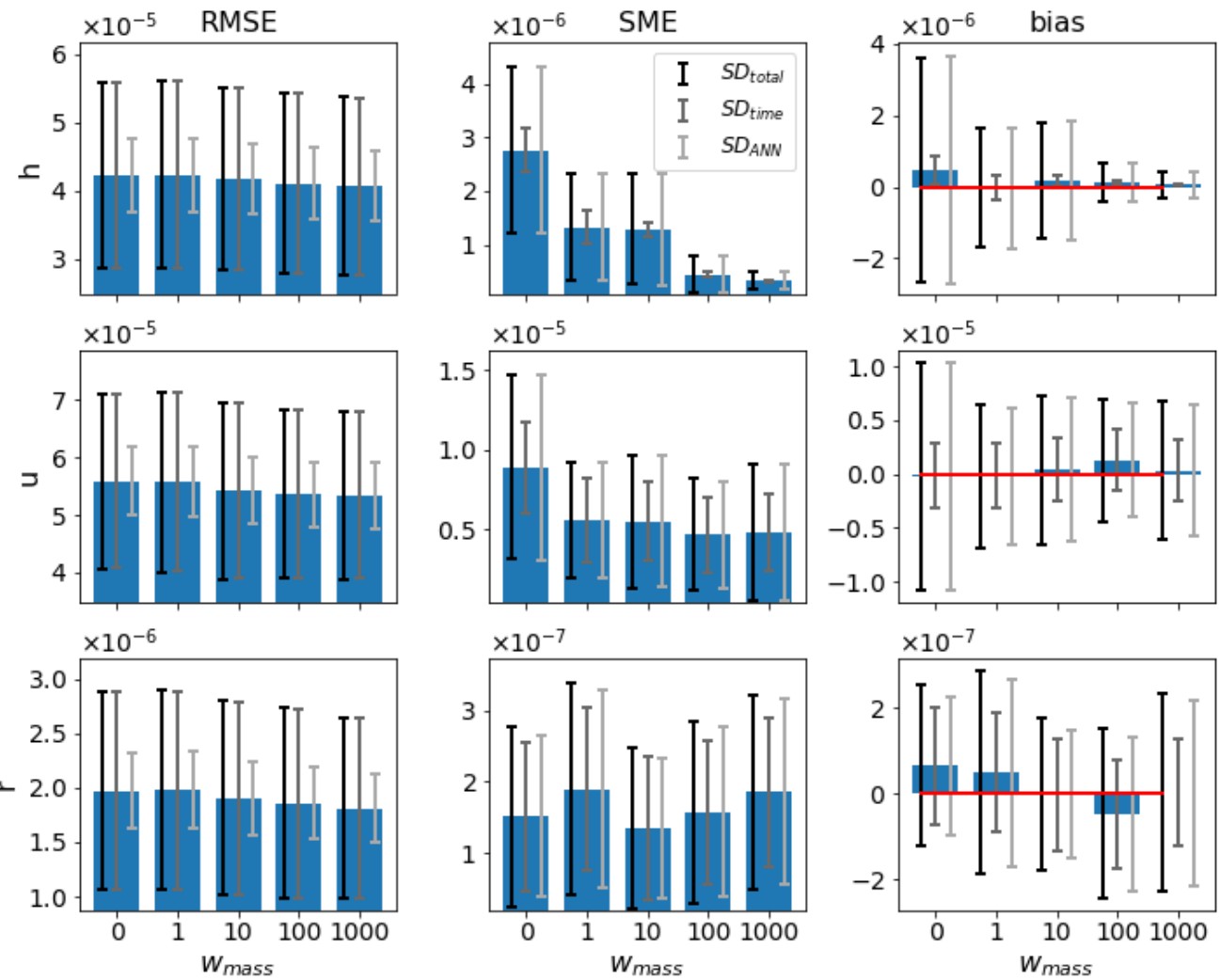

**Figure 8.** Same as Figure 7, but for the chaotic case.

## 4.2 ANN with mass conservation in a weak sense

Instead of including mass conservation in the training process of the ANN, it is natural to first try to correct the mass violation by post processing the ANN corrections. We tested two approaches: homogeneously subtracting the spatial mean of the $h$ corrections, and multiplying the vector of positive (negative) $h$ corrections with the appropriate scalar when the mass violation is positive (negative). Neither of these simple approaches led to improvements. We therefore included mass conservation in a weak sense in the training process of the ANN, as described in equation (1). We trained ANNs with mass conservation weightings of $w_{mass} = 1, 10, 100, 1000$. These weightings result in a contribution to the loss function of roughly $0.2\%, 0.7\%, 2\%$ and

5% throughout the training process respectively (not shown). Note that the ANNs presented in the previous section correspond to $w_{mass} = 0$.

Figures 7 and 8 show the single time step predictions for the regular and chaotic case respectively. Clearly, the mass conservation penalty term in the loss function has the desired effect of reducing the mass error for both orographies. Also, the error bars of the mass bias go down. A clear, convincing correlation between the reduction in SME and bias for $h$ and any other field

and/or metric is not detected, with possibly the exception of the SME for $u$ in the chaotic case. A trade-off between increasing RMSE and decreasing MSE for increasing $w_{mass}$ was expected, but is not observed. The RMSE even tends to decrease a minimal amount for the chaotic case.

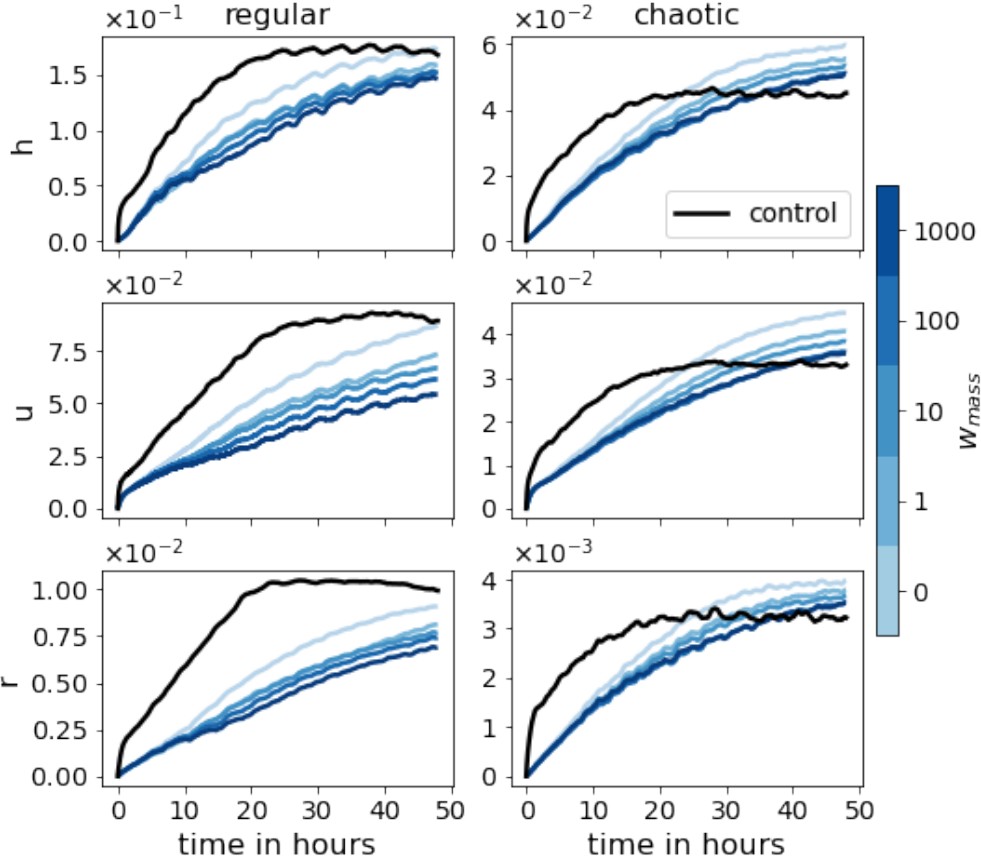

**Figure 9.** $\overline{\text{RMSE}}$ evolution of 48-hour forecasts for model variables $h$, $u$ and $r$ (from top to bottom) of $LR$ (black) and $LR_{ANN}$ for the different weightings (blues).

Figure 9 presents the mean RMSE of the 48-hour forecasts for all weightings. The weak mass conservation constraint has the desired effect on the forecast skill. For the chaotic case, more than 15 hours in forecast quality is gained. For the regular

case the number is unclear since the RMSE is still lower than $LR$ and has not yet saturated after 48 hours. However we can

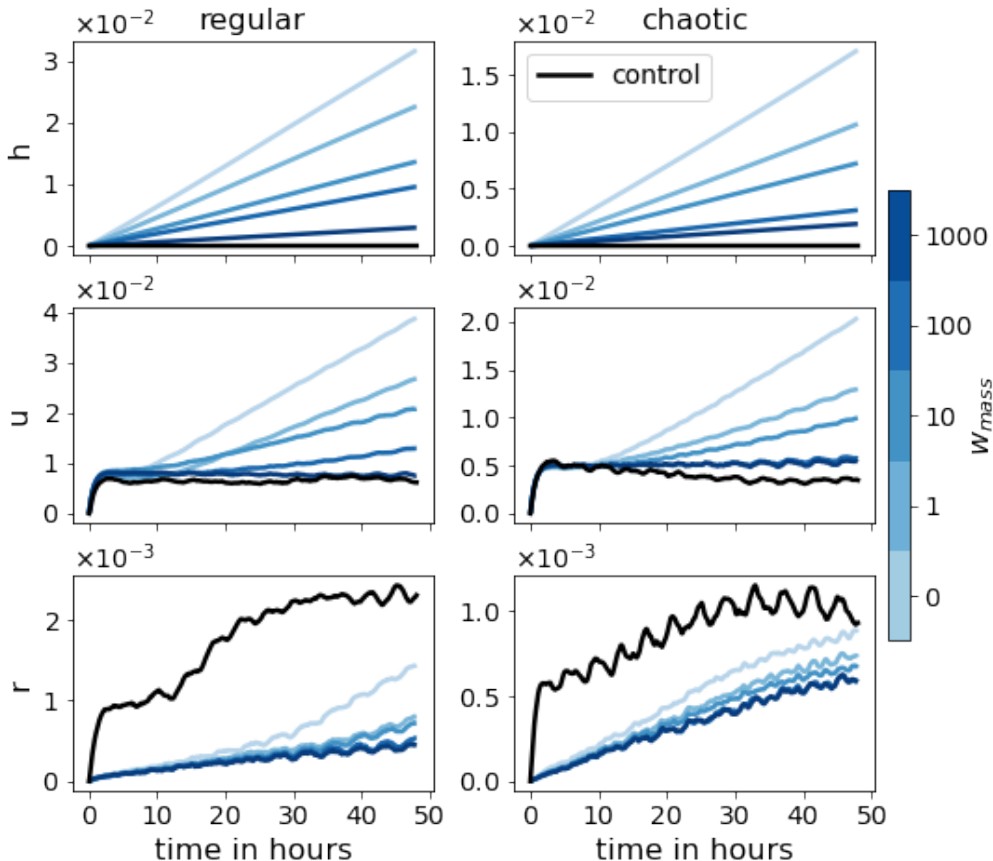

**Figure 10.** As Figure 9, but for the MSE.

say that it is at least 30 hours. As hypothesized, Figure 10 indicates that the divergence of the domain mean error of the wind $u$ is delayed as the weighting $w_{mass}$ is increased. This in turn positively affects the domain mean of the rain $r$. To support these claims we look at Figure 11, which shows the correlation between the bias in $h$ and the bias in wind $u$ and rain $r$ respectively. In the single step predictions these correlations were not conclusively detected. However, as the forecast evolves, the wind bias

becomes almost completely anticorrelated to the mass bias. A strong correlation between the mass bias and the rain bias is also established after a few time steps, likely when the change in probability of crossing the rain threshold resulting from the mass bias has taken effect. We also note that the larger $w_{mass}$, the weaker the correlations. We hypothesize that as the mass bias weakens, other causes for introducing domain mean biases in the wind and rain field become more significant. Such other causes may for example depend on the orography, or the state of $u$ and $r$.

Next we look at the variability of the forecast errors in terms of $SD_{total}$, $SD_{time}$ and $SD_{ANN}$ in Figure 12. For small weightings the variability caused by ANN realizations dominates the total variability. However, as the weighting increases, the variability

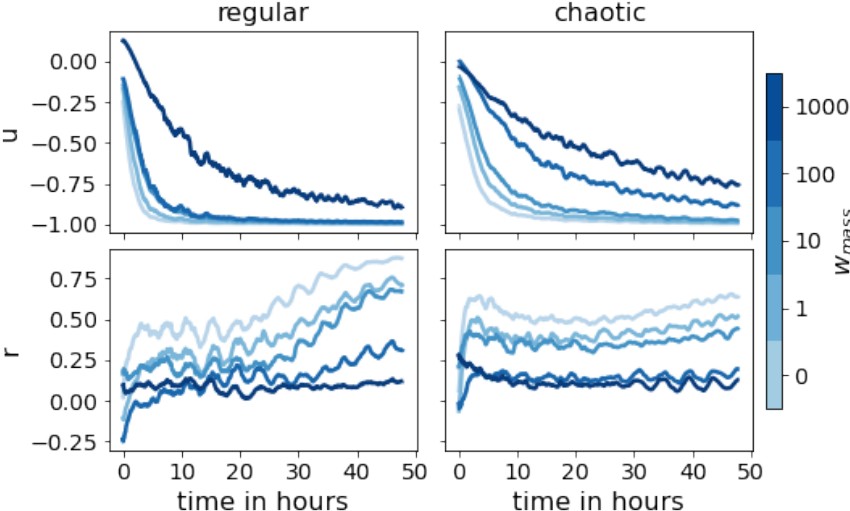

**Figure 11.** Correlation of the different weighting (blues) between the bias of $h$ and the bias of $u$ (top row) and $r$ (bottom row) for the regular (left panel) and the chaotic (right panel) case.

due to the initial conditions takes over. This again confirms the benefits of adding the mass penalty term to the loss function, as it demonstrates decrease in sensitivity of the forecast to the training process of the ANN.

A visual examination of animations of the forecast evolution suggests that convective events produced in the LR run are
235 wider and shallower than in the coarse grained HR run. This behavior mimics the collapse of convective clouds towards the grid length that is typical of km-scale numerical weather prediction models, as noted in the introduction. This then leads to a lack of rain mass, but also, via conservation of momentum, a drift in the wind field. The convective events in the LR simulations are therefore also increasingly misplaced as the forecast lead time increases. The ANNs are capable of sharpening the gradients of the convective events, leading to highly accurate forecasts of convective events up to 6-12 hours. After this,
spurious, missed and misplaced events start to occur, although the forecast skill remains significant a while longer, in contrast to the LR simulations, where the forecast skill dissolves after just a few hours. A snapshot of the state for the chaotic case is presented in Figure 13. The main rain event is misplace for $LR$ due to the bias is the wind field. Also, $LR$ misses the small neighbouring events, which the $LR_{ANN}$s do catch. Further, it is also clear that $LR_{ANN}$ for $w_{mass} = 100$ is closer to the truth for all variables then for $w_{mass} = 0$.

## 5 Conclusions

In this paper we evaluated the feasibility of using an ANN to correct for model error in the gray zone, where important features occur on scales comparable to the model resolution. The model that was used in our idealized setup mimics key aspects of convection such as conditional instability triggered by orography and resulting convective events including rain. As such, this model is representative for fully complex convective scale numerical weather prediction models and in particular the

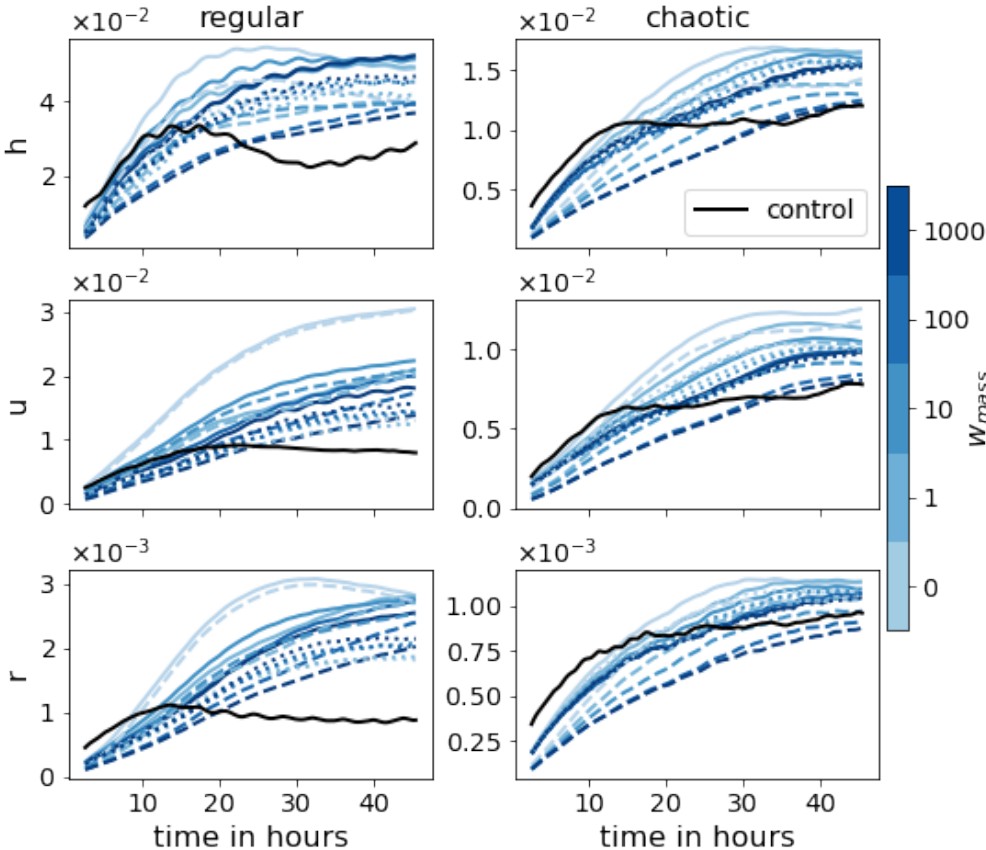

**Figure 12.** Evolution of $SD_{total}$, $SD_{time}$ and $SD_{ANN}$ (solid, dotted, dashed, respectively) for the different weightings (blues) for the regular (left panel) and the chaotic (right panel) case.

corresponding errors due to unresolved scales in the gray zone. We considered two cases, each with a different realization of the orography, leading to two different regimes. One where the convective events are large and long-lived, and one where the convective events are small and short-lived. We refer to the former case as regular and the latter case as chaotic. We showed that the ANNs are capable of accurately sharpening gradients where necessary in both cases to prevent the missing and flattening of convective events that is caused by the low resolution model's inability to resolve fine scales. For the regular case, the RMSE

is still significantly lower than the low resolution simulation ($LR$) after 48 hours. For the chaotic case, the RMSE surpasses $LR$ after about 1 day. Since the ANNs are not perfect, their errors accumulate over time, deteriorating the forecast skill. In particular, the accumulated mass error causes biases which are not present in $LR$, because the model conserves mass exactly. We therefore investigated the effects of adding a term to the loss function of the ANN's training process to penalize mass conservation violation. We found that reducing the mass error, reduces the biases in the wind and rain field, leading to further

forecasts improvements. For the chaotic case, an additional 15 hours in forecast lead time is gained before the RMSE exceeds the LR control simulation and at least 30 hours for regular case. Such positive effect of mass conservation was also found in for

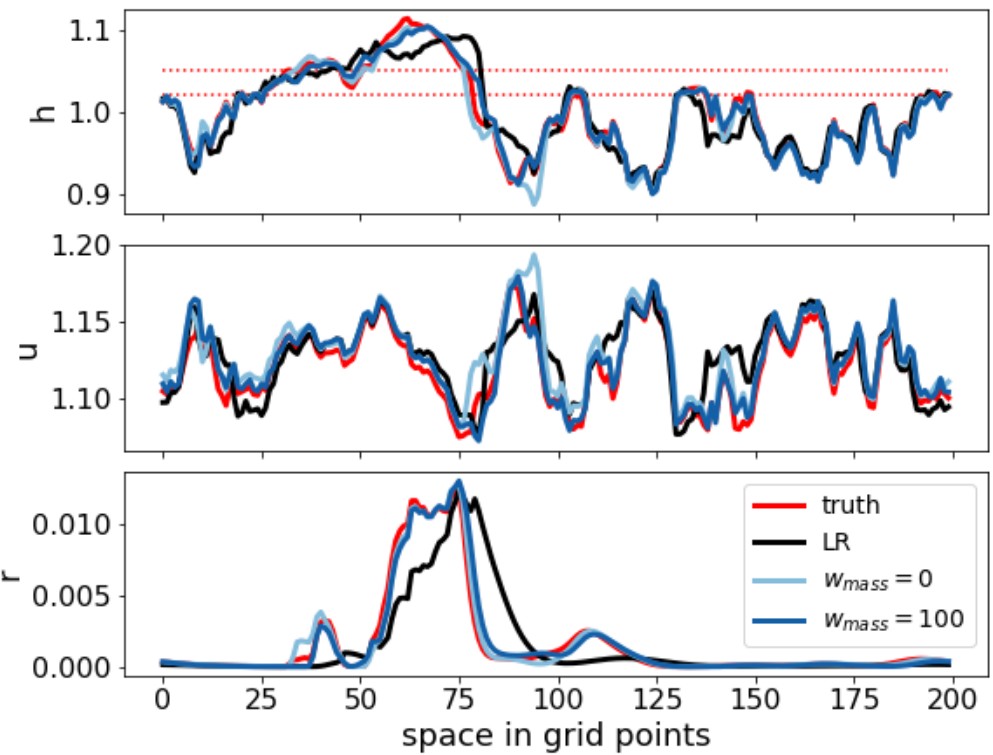

**Figure 13.** Snapshot of the state variables for the chaotic case of a 6-hour forecast starting from initial conditions of the validation data set for the truth (red), $LR$ (black), and $LR_{ANN}$ corresponding to weightings $w_{mass} = 0$ (light blue) and $w_{mass} = 100$ (dark blue). The dotted red line are the convection threshold $H_c$ and rain threshold $H_r$.

example Zeng et al. (2017); Ruckstuhl and Janjić (2018); Ruckstuhl et al. (2021). Furthermore, we showed that including the penalty term in the loss function reduces the sensitivity of the model forecasts to the learning process of the ANN, rendering the approach more robust.

While these results are encouraging, there are some issues to consider when applying this method to operational configurations. On a technical level, the generation of the training data and the training of the ANN can be costly and time consuming due to the requirement of sufficient HR data and the cumbersome exercise of tuning the ANN. The latter is a known problem that can be minimized through clever iteration of tested ANN settings, but cannot be fully avoided. Depending on the costs of generating HR data, it could be considered to use observations instead, as done by Brajard et al. (2021). They use data assimila-

tion to generate HR data from available sparse and noisy observations. Aside from saving computational costs by replacing HR simulations with data assimilation, it might offer an advantage on a different issue as well: the effect of other model error. In contrast to what was assumed in this paper, in reality not all model error stems from unresolved scales. By using observations of the true state of the atmosphere, all model error is accounted for by the trained ANN. On the other hand, the training data contains the errors inherited from data assimilation. It is not clear which error source is more important and therefore both

approaches are worthwhile investigating. Not only to improve model forecasts, but also to gain more insight in the model error itself and its comparison to errors stemming from data assimilation.

*Code availability.* The provided source code (https://doi.org/10.5281/zenodo.4740252, Kriegmair et al., 2020) includes the necessary scripts to produce the data.

*Author contributions.* RK produced the source code. RK and YR ran experiments and visualized results. SR provided expertise on neural
networks. GC provided expertise on convective scale dynamics. All authors contributed to the scientific design of the study, the analysis of the numerical results and the writing of the manuscript.

*Competing interests.* The authors declare that they have no conflict of interest.

*Acknowledgements.* The research leading to these results has been done within the Transregional Collaborative Research Center SFB/TRR 165 "Waves to Weather" (www.wavestoweather.de) funded by the German Research Foundation (DFG).

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
