# Peer review of "Using neural networks to improve simulations in the gray zone"

_Nonlinear Processes in Geophysics, 2021_

## Author Response (AR1)

**1 List of relevant changes**

- The partition of training/validation data has been changed. All plots are updated accordingly. The main conclusions remain unchanged.

- A new Figure was added showing the trajectory of both nature runs.

- A new section (section 3: Verification methods) was introduced to clarify our verification metrics.

- The bar plots for the single time step predictions (In the new manuscript Figures 4,7 and 8) are displayed differently and contain more information.

**2 Responses to Julien Brajard**

This article presents an application of a neural net to represent a parametrization of features that are not resolved by a low-resolution numerical model but that can occur at a similar scale than the low-resolution. The method is applied to the forecast of the state of a 1D shallow-water model (height, wind speed, rain mass fraction). The effect of adding a physical constraint is addressed. The article is a valuable contribution to the field of machine learning-based parametrization. It is well written, easy to follow, and the conclusions are convincing and physically interpreted. In my opinion, this work deserves publication. Nevertheless, I have 2 main comments and other secondary comments. We thank the reviewer for his important comments. These and other reviewer's comments have led to significant changes in the manuscript:

- The partition of training/validation data has been changed. All plots are updated accordingly. The main conclusions remain unchanged.

- A new Figure was added showing the trajectory of both nature runs.

- A new section (section 3: Verification methods) was introduced to clarify our verification metrics.

- The bar plots for the single time step predictions (In the new manuscript Figures 4,7 and 8) are displayed differently and contain more information.

1. about to construction of the datasets and the ANN:

- L90-91: This choice of train/validation split is very surprising. If you select one of every two points in the training, how can you be sure the training/set and the validation set are independent? On the contrary, I would expect that one state on the training is very close to the corresponding step of the validation (one time step further for example). I would be concerned about data leakage that would make you score over the validation dataset overconfident and would be unable to detect overfitting. Can you expand a bit on this choice of train/Val split?

We completely agree with reviewer and have changed our experiments accordingly. All the plots have been redone with the new data. The conclusions have not changed. We write:

"A time series of $T = 200000$ time steps, which is equivalent to approximately 57 days, is generated for both orographies. The first day of the simulation is discarded as spin up, the subsequent 30 days are used for training and the remaining 26 days are used for validation purposes. The decorrelation length scale of the model is roughly 4 hours."

- section 2.2 is there a test set? (see the following point about hyperparameters tuning)

As the learning process of our ANNs does not depend on the validation data (we don't apply any regularization such as early stopping), a test data set is less important. We also did not rigorously tune our ANNs, for which the validation data set would serve as indicator. For example, we did not change the ANN architecture and hyperparameters for our new training/validation data set. We do agree that tuning and regularization could help optimize our ANNs, in which case we would need a test data set. However, optimizing/tuning the ANNs is not the focus of this work, as this is an idealized setup.

- section 2.3 Did you need to tune the hyperparameter of the ANN (e.g. size of layers, learning rate, ...)? If so did you use the validation set or did you use a part of the training set? I think it would be nice to have a bit more details about this point...

We did loosely check out sensitivities to the architecture and hyperparameters on both the training data set and validation data set, but detected no strong sensitivities. We did not test the sensitivity to the size of the training data set. Reducing the size of the data set to the minimum required would probably increase the sensitivity to the ANN architecture. However, as argued before, optimizing the efficiency of the ANNs is not the focus of this work. We added the following sentence:

"The ANN architecture and hyperparameters were selected based on a loose tuning procedure, where no strong sensitivities were detected."

2. about physical constraint

- Eq.(1) It seems relatively "easy" to enforce strictly the water mass constraints (just remove uniformly the mean delta h at the last layer of the ANN.). Why not test this hard constraint here? First, it seems more "natural" as the weak constraint, because it is expected that the mass is strictly conserved. Second, results suggest that despite a "strong" mass constraint there is still a mass drift that makes the model diverge (Figure 9).

Good idea. We have actually tried this, but the results were not as good. We believe this is because the ANN corrects at very specific locations (depending on the state of the convection), so taking out the mass violation uniformly would also take out or add mass where the ANN did not correct anything, creating local biases. We also tried spatially splitting the ANN corrections into negative and positive corrections and multiplying the positive (or the negative, depending on the sign of the mass violation) part with the appropriate scalar to remove the total mass violation. Unfortunately this also did not lead to improvements. We added the following in the beginning of section 3.2:

"Instead of including mass conservation in the training process of the ANN, it is natural to first try to correct the mass violation by post processing the ANN corrections. We tested two approaches: homogeneously subtracting the spatial mean of the $h$ corrections, and multiplying the vector of positive (negative) $h$ corrections with the appropriate scalar when the mass violation is positive (negative). Neither of these simple approaches led to improvements. We therefore included mass conservation in a weak sense in the training process of the ANN, as described equation (1)."

Other comments:

- L27-28 "but the resolution of 2-4 km does not give accurate results for typical convective cloud structures are often less than 10 km in size". I am not a native English speaker, but this sentence is a bit unclear to me.
We have rewritten the paragraph to more explicitly the problem of poorly resolved convection in km-scale models, as follows:
"An important example of the gray zone in practice is the simulation of deep convective clouds in kilometer-scale models used operationally for regional weather prediction. The models typically have a horizontal resolution of 2-4 km, which is not sufficient to fully resolve the cumulus clouds with sizes in the range from 1 to 10 km. In these models, the simulated cumulus clouds collapse to a scale proportional to the model grid length, unrealistically becoming smaller and more intense as resolution is increased (Bryan et al., 2003; Wagner et al., 2018). In models with grid lengths over 10 km, the convective clouds are completely subgrid and should be parameterized, while models with resolution under 100 m will accurately reproduce the dynamics of cumulus clouds provided that the turbulent mixing processes are well represented. In the gray zone in between, the performance of the models depends sensitively on resolution and details of the parameterizations that are used (Jeworrek et al., 2019)."

- L71-L72 "In this study, a small but significant model intrinsic drift in the domain mean of u is accounted for by adding a relaxation term." Do you mean there is a systematic drift of u in the model? Is this at all resolutions? When it says "accounted for", does it mean that the drift is corrected?
Yes, there is a systematic drift in the model for both resolutions used in this work. Accounted for indeed means corrected in this case. We substituted "accounted for" by "removed".

- L72, you could say here that the overbar designs the domain average.
Fixed.

- L89 the index of the time t is 'i' but it was 'n' few lines above. This is still correct of course, but I feel that consistency in the notation can make the

article even clearer.

Very confusing indeed. We have fixed this.

- Table 1: Is the time step the same for the LR and HR run? Due to CFL, I would expect the time step to be smaller for HR.

Thank you for pointing this out. The output time step (that is, the time step that we save as model output) is the same for both resolutions. However, the modRSW model internally computes a dynamical model time step length at each iteration, taking into account the CFL criterion. We have added the following two sentences in training data generation section:

"The dynamical time step of the model is determined at each iteration based on the Courant–Friedrichs–Lewy (CFL) criterion. To achieve temporally equidistant output states for both resolutions, the time step is truncated accordingly when necessary. "

- L85. it is mentioned that 2 orographies are used, but I don't understand what are the 2 orographies setups here, there seems to be an ensemble of orography. It is a bit clearer in the conclusion, but I think it should also be detailed here.

We describe the generation of the orography in section 2.1 now and modified the text. We also added a Figure (Figure 1) to show the different behavior of the simulations resulting from the two orographies.

" Depending on the orography used, this model yields a range of dynamical organization between regular and chaotic behaviour. Orography is defined as a superposition of cosines with wavenumbers $k = 1/L, ..., k_{max}/L$ (L domain length). Amplitudes are given as $A(k) = 1/k$, while phase shifts for each term are randomly chosen from $[0, L]$. In this work, two realizations of the orography are selected to represent regular and more chaotic dynamical behavior. Figure ?? displays a 24 hour segment of the simulation corresponding to each orography. "

- L118: "...with the standard loss function, the MSE": maybe you could add "(wmass=0)" here (instead of mentioning it L149)

We rephrased:

"In section 3.1 we first explore the performance of the ANNs trained with the standard MSE as loss function ($w_{mass} = 0$ in equation (1))."

- L128: How do you define the $LR_{ANN}$ simulation? Is it the average of the 25 LR simulations? (Maybe this is what is meant L131, but I am not sure to understand)

To clarify this, we decided to add an entire section (verification methods), where we define more precisely the simulations that we do and how we verify them.

- L130: Initial conditions being selected every 2 hours, do you expect them to be independent? If they are not, that could bias the average and standard deviation.

We expected them to be independent, but 2 hours is indeed somewhat tight. We adjusted our experiments accordingly:

" The 48-hour forecasts are generated from a set of 50 initial conditions ($T_{veri} = 50$) taken from the validation data set. To ensure independence, the initial conditions are set 4 hours apart, which is roughly the decorrelation length scale of the model."

- Figure 4: It seems that, after around 20 hours, the dispersion of the RMSE around the mean is greater for $LR_{ANN}$ than for $LR$. Could you comment on that?

We added the following:

"Also, the $SD_{total}$ of $LR_{ANN}$ is rapidly exceeding that of $LR$. This is because, in contrast to $LR$, the shaded region for $LR_{ANN}$ includes the variability due to the ANN realizations which significantly contributes to the total variability. This is seen in Figure 12 and discussed further in the next section. "

- L155-160: Maybe it is worth mentioning that the overall effect of $w_{mass}$ on the RMSE is very low as the improvements are similar (e.g. between 97.55% and 97.70% for h, regular case)

We have rephrased this paragraph. We now say:

"A clear, convincing correlation between the reduction in SME and bias for $h$ and any other field and/or metric is not detected, with possibly the exception of the SME for $u$ in the chaotic case. A trade-off between increasing RMSE and decreasing MSE for increasing $w_{mass}$ was expected, but is not observed. The RMSE even tends to decrease a minimal amount for the chaotic case."

- Figure 10, that's a really nice point!

- L178 "Based on the subjective interpretation of the human brain of a a hand full of animations of the forecast evolution, it appears that convective events produced in the LR run are wider and shallower". Would it be some theoretical reason or literature to support this assertion?

This behavior is expected since the convergent flow narrows the convective elements until their size is close to the grid length where the collapse is stopped by numerical diffusion. A similar collapse to the grid size is typical of km-scale numerical weather prediction models, as noted in the revised introduction (see response to the first "other comment" above). The sentence at L178 has been rewritten:

"A visual examination of animations of the forecast evolution suggests that convective events produced in the LR run are wider and shallower than in the coarse grained HR run. This behavior mimics the collapse of convective clouds towards the grid length that is typical of km-scale numerical weather prediction models, as noted in the introduction."

- Figure 12: what are the dotted red lines?

We added to the caption:

"The dotted red line are the convection threshold $H_c$ and rain threshold $H_r$."

- Figure 12: Is the example taken from the training set/validation set/test set?
From the validation data set. We included this information in the caption.

**3    Responses to Davide Faranda**

I have read with interest the manuscript: "Using neural networks to improve simulations in the gray zone" by Raphael Kriegmair et al. and found it of potential interest for the public of Nonlinear Processes of Geophysics. However, before the paper could be considered for publication, I would like the authors to answer/consider the following specific comments on their work. I would be very happy to read a revised version of their paper.

We thank the reviewer for his important comments. These and other reviewer's comments have led to significant changes in the manuscript:

- The partition of training/validation data has been changed. All plots are updated accordingly. The main conclusions remain unchanged.

- A new Figure was added showing the trajectory of both nature runs.

- A new section (section 3: Verification methods) was introduced to clarify our verification metrics.

- The bar plots for the single time step predictions (In the new manuscript Figures 4,7 and 8) are displayed differently and contain more information.

Specific Comments
1) Introduction: While reading the introduction I was surprised that the authors talk about "gray zone" always avoiding mentioning the concept of turbulence (which is, by the way, mentioned in the title of one of the references provided). In my own view, and I do hope that the authors agree, the gray zone is an effect of coexisting turbulence cascades (direct and inverse) and the emergence of specific phenomena at certain scales due to the physical and geometrical constraints of the system. For exemple, in atmospheric motions, cumulus clouds and more generally convective atmospheric phenomena are constrained, in scale, by the height of the tropopopause. Similarly cyclones and anticyclones have a radius depending on Earth rotation and so on. The authors could discuss this issue and provide additional references for the gray zone with repsect to the concepts of turbulent cascades. See for exemple:

- Lovejoy, S., and D. Schertzer. "Towards a new synthesis for atmospheric dynamics: Space–time cascades." Atmospheric Research 96.1 (2010): 1-52.

- Marino, Raffaele, et al. "Inverse cascades in rotating stratified turbulence: fast growth of large scales." EPL (Europhysics Letters) 102.4 (2013): 44006.

- Faranda, Davide, et al. "Computation and characterization of local subfilter-scale energy transfers in atmospheric flows." Journal of the Atmospheric Sciences 75.7 (2018): 2175-2186.

We agree that a turbulence-based perspective on the gray zone is a useful part of the motivation of our paper, and thank the reviewer for the suggested references. We have added the following paragraph to the introduction, referring the interested reader to the excellent review of Honnert et al. (2020) for a detailed discussion.
"Viewing the atmosphere as a turbulent flow, with up- and downscale cascades, phenomena like synoptic cyclones and cumulus clouds emerge where geometric or physical constraints impose length scales on the flow (Lovejoy and Schertzer 2010, Marino et al. 2013, Faranda et al. 2018). If a numerical model is truncated near one of these scales, the corresponding phenomenon will be only partially resolved and the simulation will be inaccurate. In particular, the properties of the phenomenon may be determined by the truncation length, rather than by the physical scale. A thorough review of the gray zone problem from a turbulence perspective is provided by Honnert et al. (2020)."

2) Experiment set-up: Here the authors attempt to describe their model largely using other existing references but, even digging into the cited literature, it is complicated to understand what is the exact model used. I strongly advise to: i) write the full equations of the model (if it is too long, you can think of doing an appendix), ii) when you say "We pick one simulation from each extreme and compare results to identify general and flow dependent aspects", please show some trajectory of your model in space time (at least part of it when the system has settled in a stationary states). Figure 12 indeed shows some space snapshot of the system's stat but it comes too late in the manuscript to be useful for the casual reader.
We agree that a visual aid for the model is helpful at this point. We added the trajectory of the model in space time for both orographies as the reviewer suggests (Figure 1). The reference Kent et al., 2017 is the original publisher of the model we use and they describe the model in detail (including equations). We use their published model code. We have also made our own code available. We therefore believe it is not necessary to republish the equations.

3) Parameters used in this study: -"The coarse graining factor in this study is set to 4" why is that? the authors should provide a jutification of this value. Any reviewer or reader would question the choice of the value 4 as the only one explored in the paper. I strongly reccomend to see what happens for power-2 values, at least to some extent. In the cited paper by Faranda et al. we have seen that the coarse-grain factors can greatly affect the performances of ML methods. This item should deserve particular attention in the revision of the

paper.

It is very likely that our results will be sensitive to the choice of coarse-graining factor, as the reviewer notes, but the information gained from testing a wider range of factors is unlikely to provide useful information for the problem we are considering. Smaller factors are unlikely to be of much interest in practice, since the different resolutions are very similar. Larger factors would change the nature of the learning task by changing the physical problem, This is now discussed in the revised manuscript at L81ff.:

"The coarse graining factor in this study is set to 4, which is analogous to the range of scales found in the gray zone where deep cumulus convection is partially resolved (e.g. 2.5-10 km). Faranda et al. (2018) show that the choice of coarse graining factor can substantially affect the performance of ML methods. In our case, however, choosing a larger factor would correspond to a coarse model grid length that is larger than the typical cloud size, changing the nature of the problem from learning to improve poorly resolved existing features in the coarse simulation to parameterizing features that might not be seen at all."

-"T=200000 time steps". How can we say that this time series is long enough? what is the Lyapunov time of the system? please justify this value as, again, the length of the available dataset is a crucial parameter in ML studies.

The decorrelation length scale of the model is around 4 hours and T=200000 corresponds to approximately 57 days. We have added the following:

" A time series of $T = 200000$ time steps, which is equivalent to approximately 57 days, is generated for both orographies. The first day of the simulation is discarded as spin up, the subsequent 30 days are used for training and the remaining 26 days are used for validation purposes. The decorrelation length scale of the model is approximately 4 hours. "

-"The ANN structure used in this research is described in the following. 5 hidden layers are applied, each using the ReLU activation function. The input layer uses ReLU as well, while the layer uses a linear activation function. All hidden layers have 32 filters. The input and output layer shapes are defined by input and target data. The kernel size is set uniformly to 3 grid points." Please justify the choices "5 layers"; "32 filters" and " 3 grid points". Ideally, you should include additional tests to show that these parameters are a good choice for your analyses and why you have not attempted other combinations.

Given the amount of hyper-parameters to tune for neural networks, a dedicated hyper-parameter search would be beyond the scope of this paper. We tested slightly different configurations and did not see significant changes to the results, nor strong overfitting, giving us confidence that the NN is reasonable for the task. We added the following sentence:

"The ANN architecture and hyperparameters were selected based on a loose tuning procedure, where no strong sensitivities were detected."

4) Convolutional ANN: as for the model used, The convolutional ANN should

be defined with equations, with explicetely defined parameters. Again, if this makes the main text too long, you can move this important information in the appendix.

Since we are using the standard implementation of a convolutional neural network, a thoroughly abundant algorithm, we believe that adding equations would add a lot of text while not being useful for most readers. Rather, we now reference Goodfellow et al, a standard textbook on deep learning, that includes all the equations as used in the paper. Also, we now clarify that we use the python library Keras and list the corresponding reference. Finally, our code is made available.

5) Results: -Figure 2: 5 epochs do not seem enough to conclude anything on the variability. Why using only 5 epochs? you can use 30 and make boxplots instead of just showing 5 points. Otherwise please justify your choice 5x5

We agree that to investigate the variability stemming from epoch number versus the variability stemming from initial weights we would need more samples of each dimension. However, we are not trying to investigate the individual contribution of either of these factors. We want to sample the variability of the ANNs as a whole, for which we believe 25 samples is reasonable. Note that the computation of 50 twin experiments for each of the 250 ANNs (25 ANN realizations * 2 orographies * 5 weightings) is already a lot. The main goal of Figure 3 (new paper version, Figure 2 int the old paper version) is to justify our choice of how we get our 25 samples of ANNs. If, for example, there would barely be sensitivity to the epoch number (which is not the case), we would have been forced to train 25 ANNs for each setting to obtain our 25 samples. We rephrased:

"As the initial training weights of the ANNs and the exact number of epochs performed is to some extent arbitrary, it is desirable to measure the sensitivity of our results to the realization of these quantities. Figure 4 shows the MSE of the validation data set of the last 5 epochs (y-axis) for 5 ANNs with different realizations of initial training weights (x-axis) for both orographies. Since the MSE appears sensitive to both the initial weights and the epoch number, we use both to sample the total ANN variability, resulting in $5 \times 5 = 25$ samples for each ANN training setup that is presented in the remainder of this paper. "

-Figure 3: define RMSE

We have added section 3 "verification methods" where we define all the scores we use with equations.

-Section 3.2: it is very difficult to follow the exact way you actually train your ANN with $w_{mass}$ because you never provided the original equations. Again, my suggestion is to add the relevant equations to understand the ANN dynamics and the way you add $w_{mass}$ to improve the performances.

$w_{mass}$ only comes into play in the loss function, which is equation (1) in the manuscript. We have added references for convolutional neural networks (Goodfellow etal, 2016) and the specific python library we use (Chollet et al, 2015). In

addition we made our code available (https://doi.org/10.5281/zenodo.4740252, Kriegmair et al., 2020) .

6) Conclusions:

The authors' conclusion are consistent with the material presented in the paper. I have however suggested (see my previous comments) several way for the authors to largely improve their manuscript. In particular, I would expect to see a better model description, as well as additional analyses on the meta-paraemeters used (coarse grain factor, input layers, kernel size, and grid points numbers.